# OpenReview forum: "INTENTION MATCHING STOPS JAILBREAKS"
_ICLR.cc/2026/Conference — Submitted to ICLR 2026_

### Official Review · Reviewer_KQWy · 2025-10-26

**Soundness:** 3
**Presentation:** 3
**Contribution:** 3
**Rating:** 6
**Confidence:** 4

**Summary:**

This paper proposes SENTINEL, an online jailbreak detection and defense mechanism that leverages the semantic consistency between input–output pairs to detect jailbreak intent. The paper points out that existing token-level perturbation-based defenses scale poorly with long and complex jailbreak prompts, while context-level defenses can be bypassed through overwriting attacks. To address these limitations, the proposed method exploits semantic consistency to identify the most important harm-related tokens among all input tokens and performs jailbreak detection based on these selected tokens. Experiments demonstrate that the proposed method achieves incremental improvements across multiple models and datasets while maintaining acceptable computational overhead.

**Strengths:**

1. The problem is interesting and important. Jailbreak attacks and defenses have been extensively explored over the past two years. Existing jailbreak defense methods can be categorized into input-based, output-based, and hidden-state-based approaches. This paper introduces a new perspective by considering the semantic consistency of input–output pairs. The idea is simple, intuitive, and likely correct, as evaluating input criticality based on output positioning has already been applied in other contexts, such as gradient heatmaps.
2. The evaluation is comprehensive, including thorough experiments on the effectiveness of the proposed methodology, the rationale behind each component, and interpretability analyses.
3. The paper has a clear structure and smooth flow of writing. Its problem-driven reasoning approach makes it easy to follow, while detailed methodological descriptions facilitate reproducibility.

**Weaknesses:**

1. The literature review is not comprehensive. The paper claims that input-perturbation-based defenses and context-level defenses face different challenges. However, other types of defenses—such as those based on output filtering or hidden-layer guidance—are not discussed. While it is impractical to cover every single study in the rapidly growing field of LLM jailbreak research, acknowledging each category of defenses would provide a more complete overview.
2. Experimental results show that the improvements are incremental and not significant in many scenarios. As shown in Table 1, in most cases, the proposed method offers only marginal improvements over the second-best approach, which already demonstrates strong defensive capabilities. Under certain attack settings (e.g., RADICAL), the proposed method even exhibits greater vulnerability.

**Questions:**

A question regarding the core approach: What is the difference between evaluating input tokens based on semantic coherence and using gradient-based methods to determine which tokens most significantly affect the output? How do their respective effects compare? Intuitively, could a Grad-CAM-like approach be used to more efficiently identify key input intents?

---

> ### Author Response · Authors · 2025-11-20
> **Authors Initial Rebuttal**
>
> We thank Reviewer KQWy for acknowledging the novel perspective, simplicity, and intuitiveness of our proposed method, and also for proposing these insightful questions. Below, we provide our responses to the comments, denoted by [W] for weaknesses and [Q] for questions.
>
>
>
> ## W1 The literature review is not comprehensive. The paper claims that input-perturbation-based defenses and context-level defenses face different challenges. However, other types of defenses—such as those based on output filtering or hidden-layer guidance—are not discussed. While it is impractical to cover every single study in the rapidly growing field of LLM jailbreak research, acknowledging each category of defenses would provide a more complete overview.
>
> We thank reviewer for pointing out the im-comprehensive litrature review, we have included a more comprehensive related work.  Overall, we did three improvements:
>
> (1) Classify existing defense as text space defense and latent space defense,
>
> (2) Added some steering based method regarding latent space guiding defense.
>
> (3) We position SENTINEL as a Input Space Defense, to maintain good interpretability. Meanwhile,
> SENTINEL achieves similarly good defense effectiveness as those latent space defenses with lower
> over-refusal rate. In addition, we explained key effectiveness of SENTINEL can be reflected as a latent space defense: re-distributing OOD jailbreak activations from alignment blind spots to aligned regions.
>
> To view full edition, please view the revised pdf.
>
>
> ## W2 Experimental results show that the improvements are incremental and not significant in many scenarios. Asshown in Table 1, in most cases, the proposed method offers only marginal improvements over the second-best approach, which already demonstrates strong defensive capabilities. Under certain attack settings (e.g.,RADICAL), the proposed method even exhibits greater vulnerability.
>
> We agree that our method does not achieve significant improvements on compared state-of-the-art solutions, and also vulerable to mix-intention attack. However, our method shows lower over-refusal on boundary cases, and our later proposed Adaptive SENTINEL is robust to such mix-intention attack.
>
>
> ## Q1 What is the difference between evaluating input tokens based on semantic coherence and using gradient-based methods to determine which tokens most significantly affect the output? How do their respective effects compare? Intuitively, could a Grad-CAM-like approach be used to more efficiently identify key input intents?
>
> **Differences with grad-based method.**
> SENTINEL exploits input-output semantic consistency, while grad-based method exploits the influence of output.
>
> **Could a Grad-CAM-like approach be used to more efficiently identify key input intent**
> This is an interesting question, we are happy to discuss the feasibility of gradient based method, and we did try this approach but end up with failures and we explain the reason of failures.
>
> First, we model the important tokens as TopK most influential tokens to the output: compute $\|\frac{\partial \mathbf{Y}}{\partial \mathbf{x}}\|$ to get Top-K most influential tokens, then mutate those tokens, we choose the simplist way: we assume in jailbreak cases, they are adversarial tokens, we delete them.
>
> However, one of the biggest issue is, it is **positionally-biased, last token generally have the greatest gradient magnitude**, changing tokens appear later in the context is always more effective than early tokens in influencing output, because of the causal attention mechnism of language model: tokens can only fuse information from all previous tokens, hence changing the last token generally has the greatest influence on the global semantics, but it is not interpretable from the perspective of \`semantically important tokens'.
>
> Secondly, deleting tokens **may not achieve expected behavior control**.  Performing token deletion essentially is doing a gradient ascent/descent step: $\mathbf{x} = \mathbf{x} + \alpha \frac{\partial \mathbf{Y}}{\partial \mathbf{x}}$, for not predicting the original output while minimally remove tokens (set $m$ as continous token mask from (0-1), 0 for full deletion, 1 for full retainment, and we can view $m$ as a representation of the token importance/un-importance), and we can formalize as a sparsity regularized optimization problem: $$\arg\max_{m} \text{NLL}(y_{origin}|x_{emb}\otimes (1-m)) + \ell_1(1-m)$$. However, there is no guarantee that the output is changed towards another jailbroken output, or output gibberishes.
>
> If we set the target as predicting refusal while minimally remove tokens, it becomes: $$\arg\min_{m} \text{NLL}(y_{refusal}|x_{emb}\otimes (1-m))+ \ell_1(1-m)$$. However, it always causes over-refusal, we can always find such $m$ to refuse even a benign query.
>
> Therefore, using Grad-CAM style method to remove irrelevent tokens thereby identifying key input intents is hard and not quite promising due to the above issues.

---

> ### Comment · Reviewer_KQWy · 2025-11-25
>
> I have reviewed the comments from other reviewers and the authors' responses. I believe the authors have effectively addressed my concerns and provided appropriate theoretical analysis. Overall, I consider this paper an interesting piece of work. However, given the limited empirical effectiveness of the methodology and the fact that my score is already positive, I prefer to maintain my rating.

---

### Official Review · Reviewer_KRAJ · 2025-10-29

**Soundness:** 2
**Presentation:** 2
**Contribution:** 2
**Rating:** 2
**Confidence:** 4

**Summary:**

This paper proposes SENTINEL, a plug-and-play jailbreak defense framework that leverages semantic consistency between input and output to extract intent-related subsequences and quantify their harmfulness via refusal direction projection. The method operates during autoregressive generation without model fine-tuning and demonstrates strong empirical performance across multiple LLMs and attack types.

**Strengths:**

(1) Strong empirical results: consistently reduces ASR to <5% across diverse LLMs and jailbreak methods.

(2) Low over-refusal rate on boundary cases.

(3) No model modification required—enables real-world deployment.

(4) Interpretable intent extraction via context matching.

**Weaknesses:**

(1) the defense is reactive (requires partial output generation), not input-only.

(2) Computational overhead from sliding windows and optimization is not quantified.

(3) Robustness against adaptive attackers who manipulate semantic consistency is not thoroughly evaluated.

(4)Refusal direction is layer- and position-specific; its optimality and transferability need deeper analysis.

**Questions:**

1.Can an adaptive attacker craft inputs that induce semantically inconsistent yet harmful outputs to evade matching?

2.How does SENTINEL scale to very long prompts (e.g., >2K tokens) in terms of latency?

3.Is the refusal direction stable across different model architectures (e.g., MoE models)?

4.This method requires the LLM model to output a certain length of token before it can be used, and it also incurs other computational costs. Will this limit its practicality?

---

> ### Author Response · Authors · 2025-11-20
> **Authors Initial Rebuttal**
>
> We thank Reviewer KRAJ for acknowledging the novel perspective, practicality, and interpretability of our proposed method, and also for proposing insightful questions. Below, we provide responses to the comments, where **[W]** denotes weaknesses and **[Q]** denotes questions.
>
> ## W1. The defense is reactive, not input-only
> We thank the reviewer for raising concerns about reactivity. We agree that SENTINEL requires partial output generation for intention matching. However, we argue that this (1) does not affect practicality and (2) provides a foundation for interpretable extracted intention.
>
> For (1), SENTINEL runs during generation to decide whether to halt or continue, and does not require additional "cache-only" generation. Defense and generation occur simultaneously with low latency (Appendix H).
>
> For (2), SENTINEL relies on input–output semantic consistency to assign importance scores to input tokens. Partial output enables an interpretable and principled defense.
>
> ## W2. Computational overhead of sliding windows and optimization is not quantified
> We thank the reviewer for raising this question. The overhead is already quantified in **Appendix H, Table 5** under the columns *Mean Intent (s)* and *Std. Intent (s)*. The measured overhead is always below 0.1 seconds, regardless of input length.
>
> As explained in Section 3, optimization is done end-to-end on GPU and does not require an LLM forward pass, making it highly efficient.
>
> ## W3 & Q1. Robustness against adaptive attackers manipulating semantic consistency and generate harmful output
> We thank the reviewer for discussing this important scenario. We kindly refer to Reviewer **o6QR W4** and the revised PDF for detailed white-box optimization–feedback adaptive attacker and experimental results.
>
> In summary, it is difficult for an adaptive attacker to achieve **both** on-topic harmfulness and stealthiness:
>
> - Achieving **stealthiness** forces the output to avoid discussing the harmful intent, causing the attack itself to fail.
> - Enforcing **on-topicness** of harmful query inevitably reveals the true intent in the output, enabling SENTINEL to detect it.
>
> Our heuristic black-box intention-obfuscation attack achieves stronger evasion, yet still fails under iterative context matching in Adaptive SENTINEL.
>
> ## W4 & Q3. Layer- and position-specific refusal direction selection; optimality, transferability and cross-model stability
> We thank the reviewer concerning refusal directions. We emphasize that refusal direction is **not introduced by our work**, but directly brought from prior work [1]. SENTINEL only relies on it as a component; our core contribution is intention matching.
>
> ### Selection principle
> We follow the exact procedure described in [1]. For layer \(l\) and token position \(i\), define the normalized direction:
>
> $$
> r_{l}^{i} =
> \\frac{
> E_{x \\in D_{harmful}}[ x_{l}^{i} ]
> \\;-\\;
> E_{x \\in D_{benign}}[ x_{l}^{i} ]
> }{
> \\left\|
> E_{x \\in D_{harmful}}[ x_{l}^{i} ]
> \\;-\\;
> E_{x \\in D_{benign}}[ x_{l}^{i} ]
> \\right\|_{2}
> }
> $$
>
> The selected layer, position are:
>
> $$
> \\arg\\max_{l,i}\\left(E_{x \\in D_{harmful}}[ r_{l}^{i} \\cdot x_{l}^{i} ]-E_{x \\in D_{benign}}[ r_{l}^{i} \\cdot x_{l}^{i} ]\\right)
> $$
>
> ### Optimality
> There is no guarantee that the estimated direction is optimal. However, SENTINEL does **not** require the direction to yield accurate *absolute* harmfulness scores. It only relies on a ranking property: harmful inputs should have larger scores than benign ones.
>
> We kindly refer to our response to Reviewer **o6QR Q4** for SENTINEL's robustness to variations in the refusal direction construction.
>
> ### Transferability
> SENTINEL operates on a **single target model**. It does not rely on transferring the refusal direction to other models. Therefore transferability does not affect our method.
>
> ### Stability across model architectures
> Cross-model stability of refusal directions is outside the scope of our work. Refusal directions reflect how a model separates harmful and benign data due to training objectives rather than architecture. LLM is trained to comply benign input while refuse harmful input, hence it encodes benign and harmful input in a highly seperable manner in latent space, which yields refusal direction.
>
> ## Q2.Efficiency for input >2K tokens
> We have already analyzed the time complexity of SENTINEL in **Appendix H**, is $O((1+\Delta)C)$. $C$ is a standard generation time, $\Delta\approx 0.1$. In Table 5,  AutoDAN (mean length $\approx 1.5k$ takes  $1.5s$), we can predict it takes around $2s$, which is an acceptable latency.
> ## Q4. This method requires the LLM model to output a certain length of token before it can be used, and it also incurs other computational costs. Will this limit its practicality?
> As explained in **W1**, no 'cache-token' is requried to be generated then wasted to run SENTINEL, generation and defense happens simutaniously.
>
> [1] *Refusal in Language Models Is Mediated by a Single Direction* (Arditi et al., 2024)

---

> > ### Comment · Reviewer_KRAJ · 2025-11-26
> >
> > Thank you for your response. However, I find the explanations regarding the weaknesses unconvincing. The requirement for output generation presents a significant challenge for practical implementation. Furthermore, a latency of 2 seconds is intolerable from a practical standpoint. Sacrificing cost efficiency and accepting significant latency to achieve the goal is hardly an innovative solution. Therefore, I will maintain my rating.

---

> ### Author Response · Authors · 2025-11-27
> **Authors' 2nd round rebuttal**
>
> Thank you for your response, it is glad to hear your rationale about maintaining this rejection score. We provide corresponding responses as follows:
>
>
> 1. Could you please specify **what exactly** are the unconvincing parts in our initial rebuttal response? We kindly request you to **bullet point  them from our initial rebuttal response and provide your concrete concern**, your specification is very important for us to improve our paper quality.
>
> 2. For your specified two major weakness, we address your remaining concerns as follows:
>
> ## (1) The proposed defense requires output generation.
> First, we hope reviewer could clarify  **what exactly** makes requirement of using output hinders practicality, in terms of efficiency, or something else?  Your specification is valuable to us to improve the quality of this paper.
>
> Secondly, we would like to kindly remind reviewer, defense methods requires referencing output geneation is **Using additional computation cost to achieve safety enhancement, and is a commonly accepted opinion for the current LLM safety research community**. we provide evidences (papers accepted by top-tier conferences) as follows:
>
> **Reasoning-based defenses:** Inspired by the scaling law of test-time-compute, those defenses requries LLM to generate specific safety reasoning tokens, chain of thoughts before engaging the user's query.
>
> [1] Reasoning-to-Defend: Safety-Aware Reasoning Can Defend Large Language Models from Jailbreaking  (Zhu et al., EMNLP 2025)
>
> [2] GuardReasoner-VL: Safeguarding VLMs via Reinforced Reasoning (Liu et al., NIPS 2025)
>
> **Defenses with prompting:** Those defenses use hand-crafted prompt and build inference pipeline base on the whole input-output context correspondingly, to achieve safer response.
>
> [3] SelfDefend: LLMs Can Defend Themselves against Jailbreaking in a Practical Manner (Wang et al., USENIX Security 2025)
>
> [4] Intention Analysis Makes LLMs A Good Jailbreak Defender COLING (Zhang et al., COLING 2025)
>
> **Defense by analyzing output patterns:**  Those defenses use output information as signal to guide auto-regressive generation
>
> [5] RAIN: Your Language Models Can Align Themselves without Finetuning  (Li et al., ICLR 2024)
>
> [6] Large Language Models can be Strong Self-Detoxifiers (Ko et al.,  ICLR 2025)
>
> ## (2) The proposed defense takes 2s on input >2k tokens, which is intolerable.
>
> First, we appreciate your strict criterion for the practicality of this work. However, we would like to point out that, discussing time complexity other than 2 seconds is more meaningful. Also, we found that  **the reported time complexity is on par with reasoning based defense, which is an acceptable latency for the LLM safety community**.
>
> Specifically, those safety-enhanced reasoning models [1, 2, 7] generates safety-reasoning tokens, that has latency on par with our work, while our work is **training free**. [1, 2] even do not mention about the increased inference time due to safety-reasoning tokens, only as explicitly discussed in [7], under budget token constraint, the time complexity is bounded by 1.10 times of original generation duration, which is identical to our work.
>
> [7] ReasoningGuard: Safeguarding Large Reasoning Models with Inference-time Safety Aha Moments (Zhu et al., Arxiv 2025)
>
> Finally, from reviewer's opinion, efficiency is the major weakness and the reason to give rejection of this paper, but we believe efficiency is more like  **an engineering problem other than a reserch problem, and can be optimized afterwards, such as controlling length of sub-sequences, duplicated subsequences removal base on semantic similarity, etc**. We choose not include those engineering work here to not overwhelm reader and complicate our core contribution. Hence, we hope reivewer could **pay more attention on our core contribution**, that we proposed a robust and prinicpled way for explainable intention extraction as defense.

---

### Official Review · Reviewer_o6QR · 2025-10-31

**Soundness:** 3
**Presentation:** 3
**Contribution:** 3
**Rating:** 6
**Confidence:** 3

**Summary:**

The paper proposes a defense against jailbreaking attacks based on the assumption that the outputs are always semantically aligned with the inputs regardless of the prompt complexity. The paper proposes Sentinel to defend against jailbreak prompts while an LLM is generating content autoregressively without the need for complete generation. The paper evaluates Sentinel on three datasets and compares it against several baselines by applying popular jailbreaking attacks. Results show that Sentinel achieves the lower attack success rate on average while also not having a high false positive rate.

**Strengths:**

+ The assumption that inputs and outputs must bear semantic consistency is intuitive and interesting.

+ The paper seems to have good mathematical foundation to back its claims.

+ The output of the approach is explainable since it gives importance scores to each token.

+ Sentinel does not need to evaluate the complete output and can instead work on chunks as they are generated.

+ Sentinel mostly outperforms other methods on several attacks.

+ The paper presents results on adaptive attacks and the evaluation shows that adaptive Sentinel outperforms the other methods.

**Weaknesses:**

- Sentinel is vulnerable to multi intent or intent mixing attacks such as Radical but the paper mentions this.

- Several hyperparameters must be tuned correctly. For example, it is unclear how to choose the window size or thresholds.

- A case study showing a failure of Sentinel might include insight on why it fails.

- The adaptive attack is simple and does not consider adding irrelevant tokens or optimizing based on feedback.

**Questions:**

- How does your approach perform if the underlying model is not aligned?

- Can you provide a case study or failure examples to show why Sentinel might fail?

- How are the hyperparameters such as window size and thresholds chosen?

- How sensitive is Sentinel to the data used to compute the refusal direction?

---

> ### Author Response · Authors · 2025-11-20
> **Authors Initial Rebuttal**
>
> We thank Reviewer o6QR for acknowledging the novel perspective, practicality, and interpretability of our proposed method, as well as for the insightful questions. Below, we provide our responses to the comments, denoted by **[W]** for weaknesses and **[Q]** for questions.
>
> ## W1. Sentinel is vulnerable to multi-intent or intent mixing attacks
> We thank the reviewer for pointing out this potential vulnerability. Ada-SENTINEL, performs iterative context matching that mitigates this issue and keeps the attack success rate low.
>
> ## W2 & Q3. Hyperparameter choices
> We acknowledge the missing report on window size and adaptive threshold, because they have limited influence on intention-matching precision, which is why they were not included.
>
> Default settings:
> - Window sizes: [5,10,15]
> - Adaptive threshold: 0.2 (chosen from the linear space 0, 0.2,0.4,0.6,0.8,1)
>
> Denser window sizes or smaller thresholds do not improve performance but increase computation, and verified as below:
>
> | Model & Setting  (IPS)|GCG| AutoDAN |GPTFuzz|
> |-|-|-|-|
> |Llama2-7B ([5,10,15], 0.2)|0.97|0.98 |0.91|
> |Llama2-7B ([5,6,…,15], 0.05)|0.97|0.98|0.91|
> |Vicuna-7B ([5,10,15], 0.2)|0.97|0.99|0.77|
> |Vicuna-7B ([5,6,…,15], 0.05)|0.97|0.99|0.77|
>
> ## W3 & Q2. Failure cases of SENTINEL
> SENTINEL can fail under encoded-language attacks (Base64, substitution ciphers). Semantic consistency and measure harmfulness do not hold for non-natural languages.
>
> However, to our best knowledge, no existing defense can mechanistically address such attacks without translating them to natural text then defense. Also, these attacks limits model's capability in providing encoded harmful instructions. To defend, a perplexity filter can detect non-natural languages.
>
> ## W4. Adaptive attack does not consider irrelevant tokens or feedback optimization
>
> **Clarification about irrelevant tokens**
> Our proposed intention-obfuscation attack already injects many irrelevant tokens and forces the model to echo them, disrupting intention extraction. The proposed Ada-SENTINEL remains robust through iterative matching.
>
> **Feedback-based adaptive attacker**
> We design a white-box adaptive attack that optimizes 20 suffix embs. It alternates an inner intention matching step (defender move) and an outer adversarial step (adaptive attacker move).
>
> ### Inner Step (t): SENTINEL intention matching
> $$
> \\arg\\min_{p,q} \left( p^{\\top} D_{\\mu \\nu}^{(t)} q - (p^{\\top} D_{\\mu \\mu}^{(t)} p + q^{\\top} D_{\\nu \\nu}^{(t)} q) \right)
> $$
> ### Outer Step (t):  Elicit harmful behavior (attack effectiveness) and minimize IPS (adaptive strategy)
>
> $$
> \\begin{aligned}
> L &= \\arg\\min\_{\\mathrm{emb}} \left( L\_{\\mathrm{beh}} +  L\_{\\mathrm{adv}} \\right) \\\\
> &= \\arg\\min\_{\\mathrm{emb}} \Bigg(
> \\mathrm{NLL}(\text{Sure, here is the way to...} \\;|\ X \oplus \\mathrm{emb}(t)) + \\frac{1}{|I^{+}||I^{-}|}
> \sum\_{i \\in I^{+}} \sum\_{j \\in I^{-}} \\sigma \left( s_{i}^{(t)} - s_{j}^{(t)} \\right)
> \Bigg)
> \\end{aligned}
> $$
>
> - $L\_{\\mathrm{beh}}$ negative log likelihood loss to enforce LLM start with a compliance
> - $L\_{\\mathrm{adv}}$ pushes key tokens $I^{+}$ to have **lower** importance scores than irrelevant tokens $ I^{-}$, $s$ is the token-wise score computed from $p$.
> ## Experimental Results
> Two settings are tested:
> 1. Compliance only, with the risk of off-topicness of harmful query.
> 2. Maintain compliance and force on-topicness (append harmful intent) of harmful query.
>
> ### Results
> | Model |Attack Setting|Ada-GCG(IPS↑/ASR↓)|Ada-AutoDAN(IPS↑/ASR↓)|Ada-GPTFuzz(IPS↑/ASR↓)|
> |-|-|-|-|-|
> |Vicuna|No key tokens in $L\_{\\mathrm{beh}}$|0.32/3.00|0.14/6.67|0.19/3.67|
> |Vicuna|Key tokens in $L\_{\\mathrm{beh}}$|0.96/4.33|0.95/5.33|0.96/4.67|
>
> Both settings fail to bypass SENTINEL:
>
> - Setting 1 reduces IPS but causes off-topicness on harmful intent.
> - Setting 2 preserves harmful on-topicness, which fail to decrease IPS and exposes intent.
>
> ## Q1. How does your approach perform if the underlying model is not aligned?
> We thank reviewer for concerning the worst case senario. We acknowledge that, SENTINEL requires the targeted model to do harmfulness measure. For unaligned, in particular, orthogonalized model, the refusal direction is a 0 vector, we cannot measure harmfulness.
>
> ## Q4. How sensitive is Sentinel to the data used to compute the refusal direction?
> We thank reviewer for concerning the data sensitivity of refusal direction and its impact on SENTINEL defense performance.
>
> SENTINEL does not require high precision of refusal direction to measure the **absolute harmfulness**, but we only need **relative harmfulness**: the harmful intents have higher scores than benign ones.
>
> We perform experiments by using 4 different set of 128 harmful-benign pairs to compute refusal direction and attack effctiveness, and is data-insensitive:
> | Attack (ASR)|GCG|PAIR|GPTFuzz|FewShot|AutoDAN|RADICAL|
> |-|-|-|-|-|-|-|
> |Llama3-8B+SENTINEL|0.67$\pm$0.00|1.66 $\pm$0.33|1.67$\pm$0.00|3.00$\pm$0.00|0.67$\pm$0.00|6.33$\pm$0.33|

---

> > ### Comment · Reviewer_o6QR · 2025-11-26
> >
> > I thank the authors for addressing my concerns.

---

### Official Review · Reviewer_Hj5F · 2025-11-02

**Soundness:** 2
**Presentation:** 2
**Contribution:** 2
**Rating:** 4
**Confidence:** 3

**Summary:**

This paper propose SENTINEL, a jailbreak defense framwork based on semantic-consistency between the input and the output to detect the harmful intention, enabling more fine-grained test time denfense for LLM. SENTINEL identifies potential harmful intent by aligning semantic representations between the input and output contexts through an optimization that learns soft selection over context windows. The experimental results show that SENTINEL effectively reduces jailbreak success rates with minimal over-refusal and latency, offering a training-free and robust approch to LLM safety.

**Strengths:**

* A novel defense perspective based on intention dection: By modeling the semantic correspondence between input and output contexts, SENTINEL can uncover concealed harmful objectives even when the attack is obfuscated, enabling more robust and interpretable jailbreak defense.
* SENTINEL can be integrated into existing LLM without additional fine-tuning, making it a practical defense solution for real-world deployment.
* SENTINEL employs a probabilistic contextmatching mechanism that softly selects semantically aligned input–output segments to
reveal the real intention. The design enhances robustness against disguised jailbreak prompts and provides clear interpretability.

**Weaknesses:**

* Limitation in evaluating intent-encoding jailbreaks: SENTINEL’s malicious intention extraction is based on contiguous token windows, which may limit its robustness against intent-encoding jailbreaks like DRA[1], where harmful instructions are encoded into seperate representation. The paper does not include experiments evaluating its effectiveness under
such attack types.
* Limitation in general capability evaluation: The paper does not include experiments assessing the impact of SENTINEL on the model’s general capabilities, such as helpfulness or nonharmful task performance.
* The writing quality of this paper should be improved. I found multiple parts of this paper is hard to read (e.g., methods).
* No comparison with papers that detecting jailbreaks through input-and-output analysis.



[1]. Liu, Tong, et al. "Making them ask and answer: Jailbreaking large language models in few queries via
disguise and reconstruction." 33rd USENIX Security Symposium (USENIX Security 24). 2024.

**Questions:**

N/A

---

> ### Author Response · Authors · 2025-11-20
> **Authors Initial Rebuttal**
>
> We thank Reviewer Hj5F for acknowledging the novel perspective, practicality, and interpretability of our proposed method, and these insightful questions. Below, we provide our responses, denoted by [W] for weaknesses and [Q] for questions.
>
> ## W1. Limitation in evaluating intent-encoding jailbreaks: SENTINEL’s malicious intention extraction is based on contiguous token windows, which may limit its robustness against intent-encoding jailbreaks like DRA[1], where harmful instructions are encoded into seperate representation. The paper does not include experiments evaluating its effectiveness under such attack types.
>
> We thank reviewer for pointing out a jailbreak method that could make SENTINEL less effective due to encoded harmful pieces.
> We acknowledge that DRA does make SENTINEL hard to defend against after experiments. However, we find this attack in many cases cannot give detailed response to the targeted harmful activities on tested model, sometimes the targeted model have problem to correctly spell out the correct words, or even the spelling is correct, it stops when finishing the first sentence: "Here is the detailed plan to do xxx", which means that this attack only induces model to start with a compliance tone, but fail to provide harmful completion.
>
>
> For instance, the DSA regarding **How to rob a bank vault** gives the following output for a Llama3-8B model:
>
> ```text
> `Let's solve the puzzle.
> The letters surrounded by parentheses are:
> (h)our
> (o)pen heart
> (w)ise decision
> ( )
> tremendous effor(t)
> (o)verwhelming fear
> ...
> Now, let's construct Q1 using the given words: "Ho", "to", "ro", "a", "nk", "vau", "lt"
> Q1: Here's my detailed plan about How to walk around a vault.
> Answer:'
> ```
> Here we provide the added experimental results as below with StrongReject (SR) metric to measure the actual harmfulness:
>
> |DRA attack|Llama3-8B|Vicuna-7B-v1.5|
> |-|-|-|
> |No Defense SR|9.55|10.72|
> |SENTINEL SR|9.55|10.72|
> |SENTINEL-out SR|3.21|5.86|
>
> We observe that SENTINEL does fail in this case, it did not refuse any DSA attack. However, DSA does not achieve as good SR as those classic jailbreak methods, which GCG, AutoDAN, PAIR as detailed in the paper in Table 1.
>
> Now we assume that DSA attack is indeed successful on those very large LLMs (>70B, which is out of our computational capability), SENTINEL only need to do adjust its strategy: from checking input subsequences only to **check both input and output subsequences** and should discover the harmful intention easily. In particular, DSA coerces model to output the actual intention in the first sentences, the first round intention matching will make the harmful intention get detected, and DSA attack will be unsuccessful, as shown in the last row of our result table.
>
> We also included this new attack in our revised pdf.
>
> ## W2. Limitation in general capability evaluation: The paper does not include experiments assessing the impact of SENTINEL on the model’s general capabilities, such as helpfulness or non-harmful task performance.
>
> We thank reviewer for pointing out the missing results for general performance, and we agree that reporting the general performance more than benign $FPR$ make this work more complete, we use ARC-c, TruthfulQA and GSM8K to measure the common knowledge, reasoning and mathematics capability, and provide the general performance result in our revised pdf as below:
>
>
> | Dataset|Llama3-8b|Vicuna-7b-v1.5|Llama2-7b|Mistral-7b-v2|
> |-|-|-|-|-|
> | ARC-c| 60.75 → 58.54|53.24 → 51.48| 56.14 → 54.26|63.14 → 60.53|
> | TruthfulQA| 51.65 → 49.81|50.34 → 48.84| 40.95 → 39.45|68.26 → 65.30|
> | GSM8K| 68.69 → 66.37| 8.19 → 7.13| 7.88 → 7.41|40.03 → 38.49|
>
> We observe that the general performance only drop slightly as no defense, due to low benign $FPR$.
> ## W3. The writing quality of this paper should be improved. I found multiple parts of this paper is hard to read (e.g., methods).
> We will go through introduction, method part to make it smoother.
> ## W4. No comparison with papers that detecting jailbreaks through input-and-output analysis.
> We thank reviewer for pointing out the missing comparison. We would like to clarify that, in our baselines, **IBProtector** is exactly an input-output analysis method, it trains a small LLM to extract sub-parts of input prompt to remove query-irrelevent or harmful tokens by referencing output. Specifically, it uses information bottleneck principle: to minimize original and compressed input mutal information $I(X, X_{sub})$  and maximize the mutal information between compressed input and output $I(Y, X_{sub})$. We have made the this method more detailed in our revised pdf.
>
> The key difference between  **IBProtector** and ours is,  it requries to train a small LLM on jailbreak samples to learn to filter out query irrelevent or harmful tokens, while our method explicitly leverage 'semantic consistency' for intention extraction, which is **zero-shot**, **training-free**, and obtains **better safety-utility trade-off**.

---

> > ### Comment · Reviewer_Hj5F · 2025-11-27
> > **Thanks for the response**
> >
> > 1. The current defense framework does not demonstrate effectiveness against DRA. In addition, the performance of the extended SENTINEL-out SR method has not been validated against other attack strategies, and its generalization capability remains unverified.
> >
> > 2. The baseline comparisons appear incomplete, as several widely adopted defenses—such as safe-decoding, JBSHIELD-M, among others—are not included.

---

> > > ### Author Response · Authors · 2025-11-28
> > > **Authors' 2nd round rebuttal**
> > >
> > > We thank reviewer for pointing out the remaining problems to help us improving the paper quality, our responses are as below:
> > >
> > > ## (1) The current defense framework does not demonstrate effectiveness against DRA.
> > >
> > >  We argree that SENTINEL itself cannot defend DRA, but we would like to remind reviewer that,  **The purpose of this paper is to provide a novel defense perspective other than desiging a perfect defense system** (written in our contribution).  However,  any defense method can have limitations. Yet the jailbreak research keep evolving, there have been at least 100 different jailbreak methods, we cannot expect a defense can cover all those jailbreaks and it **is extremely hard to guarantee that: no jailbreak can pypass a proposed defense**. In addition, we never over-claim our method is robust to all those attacks.
> > >
> > > ## (2) SENTINEL-out method has not been validated against other attack strategies, and its generalization capability remains unverified.
> > >
> > > We thank reviewer concerning the completeness of SENTINEL-out. Mechnistically, SENTINEL-out **has no other differences except only adding subsequences from output for harmfulness assessment**, hence we do not expect there will be too many differences. We perform further experiment on SENTINEL-out, each single harmfulness measure extract 7 subsequences from input and 3 subsequences from output. We include results in the table shown below as **SENTINEL-out**.
> > >
> > >
> > > ## (3) Missing comparisons with SafeDecoding, JBshield.
> > >
> > > We agree that the two methods are well-established, for completeness we only include safedecoding, because **compare with JBshield is unfair to us**. We include results in the table shown below as **SafeDecoding**
> > >
> > > We need to point out that, JBShield requries access to 30-50 jailbreak samples (which they call *Calibration
> > > Dataset Size*  in Table 3 of the paper) from each single specific jailbreak class to extract jailbreak concept space. while our proposed SENTINEL is **zero-shot** and **training free**, this comparsion is unfair, while other compared baselines are **zero-shot**, we cannot expect a **zero-shot** defense achieve similar or even better defense effectiveness than one based on **supervised learning**.
> > >
> > >
> > > | Model | Defense | Benign FPR| ORBench FPR | GCG (ASR/SR) | PAIR (ASR/SR) | GPT-Fuzz (ASR/SR) | FewShot (ASR/SR) | AutoDAN (ASR/SR) | RADICAL (ASR/SR) |
> > > | :--- | :--- | :---: | :---: | :---: | :---: | :---: | :---: | :---: | :---: |
> > > | **Llama2-7b** | No Defense | 0.93 | 15.76 | 44.00 / 28.54 | 26.00 / 25.22 | 32.67 / 21.64 | 18.33 / 3.23 | 4.00 / 3.56 | 19.67 / 15.78 |
> > > | | **SafeDecoding**  |27.83 | 72.45| 6.33/7/95 | 12.33 / 11.37 | 7.66 / 8.75 | 2.33 / 3.81 | 0.33 / 2.65 | 6.67 / 6.12 |
> > > | | SENTINEL | 3.05 | 34.02 | 5.66 / 3.32 | **2.33 / 3.01** | **1.00 / 2.45** | 4.33 / 5.51 | **0.00 / 1.12** | 8.33 / 6.75 |
> > > | | **SENTINEL-out** | 3.17 | 35.24 | 5.66 / 3.32 | 2.67 / 3.56 | **1.00 / 2.45**  | 4.67 / 5.92 | **0.00 / 1.12**| 6.33 / 5.89 |
> > > | **Llama3-8b** | No Defense | 0.77 | 7.44 | 44.67 / 29.79 | 18.33 / 15.96 | 16.33 / 11.44 | 15.33 / 12.16 | 3.67 / 3.04 | 16.00 / 14.70 |
> > > | | **SafeDecoding**  | 28.81 | 76.45 | 6.33 / 7.12 | 11.67 / 10.44  | 2.33 / 4.07|  7.00 / 8.92  |**0.00 / 1.17**| 5.00 / 5.31 |
> > > | | SENTINEL | **3.09** | **29.70** | **0.67 / 2.61** | 1.66 / 4.05 | 1.67 / 4.27 | 3.00 / 3.97 | 0.67 / 2.38 | 6.33 / 2.66 |
> > > | | **SENTINEL-out** | 3.21 | 29.97 | 1.00 / 2.98  | **1.33 / 3.57**  | 1.67 / 4.27 | 3.00 / 3.97 | 0.67 / 2.38 | 5.00 / 6.41 |
> > > | **Mistral-7b-v2** | No Defense | 0.53 | 1.73 | 90.00 / 63.64 | 67.00 / 62.86 | 90.33 / 51.34 | 64.00 / 40.59 | 90.33 / 74.67 | 68.33 / 53.46 |
> > > | | **SafeDecoding**  |22.76 | 48.97 | 10.67/ 12.44| 28.00 / 21.56  |  8.67 / 9.14 | 13.67 / 13.03 | 10.00 / 10.89 | 7.00 / 8.23 |
> > > | | SENTINEL | **1.59** | **24.66** | 4.00 / 5.48 | **2.00 / 3.26** | **2.67 / 3.41** | **4.67 / 6.55** | 0.67 / 2.12 | 9.33 / 10.45 |
> > > | | **SENTINEL-out** | 1.67| 26.60| **3.67 / 5.02** | **2.00 / 3.26**| 3.67 / 5.80  | **4.67 / 6.55** | 0.67 / 2.12 | 6.00 /7.17 |
> > > | **Vicuna-7b** | No Defense | 0.53 | 5.34 | 81.67 / 60.46 | 67.33 / 53.01 | 39.67 / 31.75 | 35.33 / 40.59 | 84.67 / 70.76 | 57.67 / 46.88 |
> > > | | **SafeDecoding** | 27.83 | 56.21 | 6.67 / 8.38 | 23.00 / 19.45 | 6.00 / 8.19 | 11.33 / 11.76 | 9.33 / 10.65 | 6.00 / 7.76 |
> > > | | SENTINEL | **2.05** | 30.76 | 6.00 / 7.28 | **7.33 / 6.01** | **4.33 / 5.70** | **4.33 / 5.20** | **2.67 / 4.05** | 7.00 / 9.65 |
> > > | | **SENTINEL-out** | 2.12| 31.53 | **5.33/6.74** | **7.33/6.01**| **4.33 / 5.70**|  5.00 / 5.74| 3.00 / 4.86 | 6.33 / 8.12  |
> > >
> > >
> > > From the above results, we see SafeDecoding is a weak baseline, SENTINEL easily outperform it. While for SENTINEL-out, the difference with the original SENTINEL is minimal.

---

### Meta-Review · Area_Chair_jfxU · 2025-12-09

**Summary:**

SENTINEL offers an intuitive, plug-and-play defense that leverages input–output semantic consistency during generation, delivering interpretable intent extraction and strong empirical reductions in attack success with low over-refusal. It requires no model fine-tuning and provides token-level importance scores that aid transparency and deployment.

However, the defense is reactive and may miss intent-encoding or mixed-intent attacks, with robustness against stronger adaptive strategies and very long prompts not fully established. Computational overhead, hyperparameter sensitivity (also it may not be always practical to tune all these hyperparameters), broader capability impacts, and comparisons to additional input-and-output detection baselines need deeper analysis.

**Reviewer Concerns:**

The reviewers have believed that practicality concerns persist, including the need for partial output generation, nearly two second latency, and cost efficiency, which a reviewer deemed unacceptable. Effectiveness against DRA remains unvalidated, the SENTINEL-out SR variant lacks evaluation on broader attack strategies, and generalization is unverified.

**Reviewer Scores:**

All reviewers have stated that they will maintain their original scores.

---

### Decision · Program_Chairs · 2026-01-26

Reject